# The Protective Effects of Moisturizer Containing *Potentilla anserina* Extract in the Topical Treatment of Skin Damage Caused by Masks

**DOI:** 10.3390/ijms241814294

**Published:** 2023-09-19

**Authors:** Hyeong Choi, Ji Hoon Ha, Hee Cheol Kang, Won Sang Seo, Bum-Ho Bin

**Affiliations:** 1Department of Applied Biotechnology, Ajou University, Suwon 16499, Republic of Korea; hyeongchoi@ajou.ac.kr; 2R&D Complex, Kolmar Korea, Seocho 06800, Republic of Korea; jh_cos@kolmar.co.kr; 3Green & Biome Customizing Laboratory, GFC Co., Ltd., Hwaseong 18471, Republic of Korea; michael@gfcos.co.kr

**Keywords:** COVID-19, mask, redness, skin hydration, skin moisture, TEWL, *Potentilla anserina* extract, hydroxypropyl-β-cyclodextrin, liposome

## Abstract

The use of face masks during the COVID-19 pandemic resulted in significant societal changes, particularly for individuals with sensitive skin. To address this issue, the researchers explored traditional medicine and identified *Potentilla anserina* extract as a potential solution due to its anti-inflammatory and moisturizing effects. This research investigated how this extract influences skin hydration, barrier function, and itching. The findings revealed that the extract had a hydrating effect by elevating Aquaporin-3 (AQP3) expression. Additionally, the study demonstrated that the extract improved skin barrier function, with Filaggrin (FLG) expression being approximately three times higher (*p* < 0.001) in the *Potentilla-anserina*-extract-treated group compared to the control group and the genes associated with itching being reduced. In this process, we researched and developed HPβCD (hydroxypropyl-β-cyclodextrin)-Liposome containing *Potentilla anserina* extract, gradually and sustainably releasing the active components of the *Potentilla anserina* extract. During four weeks of clinical trials involving individuals wearing masks for over 6 h a day, a moisturizer containing *Potentilla anserina* extract demonstrated a notable reduction in skin redness. Hemoglobin values (A.U.), which serve as indicators of skin redness, showed decreases of 5.06% and 6.74% in the test area inside the mask after 2 and 4 weeks, respectively, compared to the baseline measurements. Additionally, the moisturizer containing *Potentilla anserina* extract notably decreased Trans Epidermal Water Loss (TEWL), with reductions of 5.23% and 9.13% observed in the test area inside the mask after 2 and 4 weeks, respectively. The moisturizer, especially in the test area treated with the extract-containing moisturizer, significantly enhanced skin hydration compared to the control group. The Corneometer values (A.U) exhibited notable increases of 11.51% and 15.14% in the test area inside the mask after 2 and 4 weeks, respectively. These discoveries emphasize the potential of *Potentilla anserina* extract and its utility in tackling skin issues caused by mask wearing, including enhancing moisture, fortifying the skin’s barrier, and alleviating itching. These results indicate that moisturizers incorporating specific ingredients provide greater benefits compared to conventional moisturizers.

## 1. Introduction

The emergence of COVID-19 caused a major crisis for modern human civilization, similar to the outbreaks of SARS in 2002, H1N1 influenza in 2009, and MERS in 2012 [1], The durations of the pandemic outbreaks appear to be decreasing in a cyclical pattern. A significant shift in behavior is the widespread adoption of wearing face masks, which has become a crucial personal and public health measure. However, wearing masks for extended periods of time may result in skin issues such as acne, contact dermatitis, erythema, papules, and pustules on the face [2,3,4,5]. The human skin, being one of the body’s largest and most intricate organs, performs a vital role in maintaining overall balance and health. It acts as a protective barrier against environmental factors, helps to regulate hormones and chemicals, and plays a role in the aging process [6,7]. Prolonged mask wearing can affect these functions. There has been a growing interest in studying the impact of masks on skin, especially sensitive skin, as concerns about skin changes caused by masks have become more widespread [8,9]. Previous studies have investigated the effects of masks on skin [10], but their impact on entire sensitive skin remains unclear. The idea that masks will no longer be necessary after COVID-19 ends is wrong. This is because an unusual increase in respiratory syncytial viruses at the beginning of the winter season was observed after masks were taken off [11]. As the speed of respiratory virus outbreaks increases, it has become difficult to imagine a world without masks.

The first study on sensitive skin was conducted in the UK in 2001, defining it as skin with a reduced tolerance to environmental factors such as heat, wind, cold, UV light, cleaning agents, cosmetics, water, and stress [12,13]. Although there are research results indicating that moisturizers can alleviate the irritation caused by masks [10], there are no studies that have improved not only irritation, but also itching and the skin barrier. There have been reports that the prolonged use of masks weakens the skin barrier [14], and individuals who wear masks for extended periods are prone to experiencing greater levels of itching [15], but there is not much known about specific substances and methods, aside from simply applying moisturizer to skin. Until now, most research has focused on improving mask irritation by just using simple moisturizers [10,14]. Skin irritation presents a complex challenge, and relying solely on moisturizers has its constraints. Additionally, moisturizers can occasionally cause discomfort, especially for those with sensitive skin, with the primary adverse effect being skin irritation, which may or may not exhibit visible signs of inflammation [16]. Addressing a wide range of skin issues, including skin barrier breakdown and irritation, is essential. Therefore, there is a societal need for the discovery of substances capable of addressing these functions.

*Potentilla* is a genus of flowering plants from the Rosaceae family that has been used as traditional medicine for a long time. There are various species, such as *Potentilla erecta*, which was used to treat oral cavity ulcerations and purulent facial eczema by a Greek physician, and *Potentilla paradoxa* Nutt. (also called *Potentilla supina* L.), a traditional medicinal herb from India, which has been used to treat astringent and febrifuge [17]. A previous study found that an ethanol extract of *Potentilla paradoxa* Nutt. had anti-inflammatory effects [18], could also promote the expression of hyaluronic acid synthesis-related enzymes, and suppress the expression of hyaluronic acid degradation-related enzymes [19]; this means that the *Potentilla* genus is also effective for moisturizing skin.

*Potentilla anserina* has a history of over 1500 years of use as both a medicine and food [20]. In Tibet, an extract of *Potentilla anserina* was used to treat viral infections [21], and it has also been historically significant for its medicinal properties [21]. *Potentilla anserina* also has several physiological benefits, such as safeguarding the liver, enhancing the functions of the spleen and the stomach, and exhibiting anti-tumor properties [20]. While close relatives, *Potentilla paradoxa* Nutt., have been reported to have skin-related benefits [17,19], there is very limited documentation regarding the skin-related effects of *Potentilla anserina*.

This experiment aimed to assess the effectiveness of *Potentilla anserina* extract at the cellular level of the skin concerning the issues triggered by mask wearing. Most importantly, it sought to establish an efficient system for delivering this extract into the skin. Lastly, the experiment aimed to verify that moisturizers containing specific extracts, rather than mere moisturizer application, alleviate the redness and itching caused by masks by enhancing the skin barrier.

## 2. Results

### 2.1. Confirmation of the Presence of Caffeic Acid in Potentilla anserina Extract

In selecting the target compounds within the extract, we focused on a report indicating the presence of caffeic acid in *Potentilla anserina* extract [22]. Caffeic acid has been shown to have anti-inflammatory, immunomodulatory, and antioxidant effects and to suppress lipid peroxidation [23,24]. As shown in Figure 1A, the presence of caffeic acid in the extract was confirmed. To verify if this fraction indeed contained caffeic acid, UV spectrum (Figure 1B) and LC-MS (Figure 1C) analyses were conducted, revealing its similarity to caffeic acid. Therefore, caffeic acid was used as a standard for percutaneous absorption and a following formulation was developed.

### 2.2. The Effect of Potentilla anserina Extract on Aquaporin-3 (AQP3) Increase

Aquaporin-3 (AQP3) is a protein that plays a role in facilitating the movement of water and glycerol across the cell membranes [25]. Treating HaCaT cells with the *Potentilla anserina* extract showed an increase in the expression of AQP3, a protein that plays a crucial role in skin hydration by allowing the movement of water into cells (Figure 2A). Compared to the control group, the sample treated with the extract demonstrated a statistically significant increase in the expression of AQP3 in a dose-dependent manner. These results suggested that the *Potentilla anserina* extract may have effectively increased AQP3 expression, leading to improved skin hydration.

### 2.3. The Effect of Potentilla anserina Extract on Filaggrin (FLG) Increase

Filaggrin (FLG) is a crucial component of the skin’s protective barrier against external factors, and its deficiency is strongly linked to conditions such as atopic dermatitis [26]. In this study, the effect of the *Potentilla anserina* extract on the expression of FLG, which is associated with skin barrier improvement, was investigated. The results indicated that the expression of FLG was about three times higher (0.05%, *p* < 0.001) in the group treated with the *Potentilla anserina* extract compared to the control group (Figure 2B). These findings suggest that *Potentilla anserina* extract can effectively increase the FLG expression and enhance the skin barrier.

### 2.4. The Effect of Potentilla anserina Extract on Itching-Related Cytokines

Cytokines such as Thymic Stromal Lymphopoietin (TSLP), Interleukin-33 (IL-33), and Interleukin-4 (IL-4) are commonly expressed in atopic dermatitis and play roles in causing skin itching [27,28,29]. IL-33 is a protein that plays a crucial role in amplifying the body’s immune response, particularly the type 2 immune reaction, and is an important target for the treatment of dry skin pruritus and chronic pruritus of an unknown origin [27]. TSLP induces TH2-mediated inflammation and boosts the release of periostin from keratinocytes, which, in turn, enhances the signaling of itching [28]. The activation of sensory neurons by IL-4 is a significant contributor to the development of chronic itch, which is a disruption of the protective response. In mice and humans, a small group of sensory neurons express the IL-4 receptor and exhibit calcium responses when treated with IL-4 in culture. These neurons are thought to be involved in the mediation of itching [29]. In the experiment, the treatment of the *Potentilla anserina* extract on keratinocytes led to reductions the expressions of the TSLP, IL-33, and IL-4 genes, indicating that the extract may have had a notable effect on reducing itching (Figure 2C–E).

### 2.5. Physicochemical Characteristics and Sustained Release of HPβCD-Liposomes

HPβCD-Liposome is known for its ability to regulate drug release rates through sustained release, thereby reducing skin irritation [30]. To investigate the physicochemical characteristics of *Potentilla anserina* extract-in-HPβCD-in-liposome and its sustained release effects, we encapsulated caffeic acid (CA), a major component of *Potentilla anserina* extract [22], in HPβCD-Liposome and evaluated its physicochemical characteristics and sustained release effects. The particle size of the HPβCD-Liposome containing caffeic acid (F2) was approximately 215.2 nm, which was larger than the particle sizes of the caffeic acid liposome (F1) and empty liposome (F0), which were 201.3 nm and 195.2 nm, respectively (Table 1). The encapsulation of caffeic acid and HPβCD in liposomes resulted in an increased particle size. The zeta potential of F0 was −28.2 mV, which was lower than that of F1 and F2. The inclusion of caffeic acid and HPβCD may have contributed to an increased zeta potential, enhancing stability. The encapsulation efficiencies (EE) of CA in F1 and F2 were 82.5% and 75.6%, respectively. This suggests that CA, which had an increased solubility due to its inclusion in HPβCD, may not have been completely encapsulated in the liposome core and dispersed in the liposome’s aqueous phase [31].

In terms of the CA release effects, F1 exhibited a slight and sustained release of CA, reaching 51% at 4 h, followed by a slower release rate, reaching 55% at 6 h (Figure 3A). In contrast, F2 showed a higher initial CA amount, measured at 5%, and a faster release of CA, at approximately 26% within the first hour, compared to F1. This suggested the presence of CA/HPβCD in the external phase due to incomplete encapsulation in the liposomes during manufacturing, leading to a rapid CA release. However, after 1 h, the release rate of CA in F2 decreased, resulting in a lower cumulative release of CA at 2 h compared to F1. Subsequently, the CA release rate in F2 remained relatively constant. These results indicated that the structure of the CA encapsulated in both HPβCD and liposomes exhibited a sustained release compared to the CA solely encapsulated in liposomes. It was expected that the *Potentilla anserina* extract would follow a similar release pattern.

### 2.6. Skin Permeation Effects of HPβCD-Liposome

To investigate the skin permeation effects of HPβCD-Liposome, we compared the liposomes encapsulating CA (F1) and HPβCD and liposomes (F2) to a control group of CA dissolved in 1,3-BG using Franz cells (Figure 3B).

At 1 h after the sample treatment, F1 and F2 showed similar results, but they exhibited approximately twice as much CA penetration into the skin compared to the control group. At 2 and 4 h, F2 showed significant increases of 4.2 and 1.4 times, respectively, compared to the control group and F1. Furthermore, from 2 to 12 h, the skin penetration of CA by F1 and F2 consistently increased, unlike that of the control group. F2 exhibited approximately a 1.5 times higher skin permeation compared to F1, suggesting that HPβCD provided some assistance. Simply dissolving CA in a solvent is challenging for skin penetration due to the hydrophobic nature of the component. In contrast, the liposomes increased the skin permeation effect of CA through their skin affinity and elasticity. The enhanced skin delivery effect of HPβCD-Liposome helped the active components of the *Potentilla anserina* extract to penetrate the skin rapidly.

### 2.7. Improvement Effects Regarding Skin Redness

Due to a report that skin moisturization decreases and skin redness increases due to the use of masks, we selected participants (33 subjects) who used masks for more than 6 h per day for the test. A total of 33 subjects were recruited and a four-week clinical trial was conducted to evaluate the moisturizing and barrier-improving effects of a moisturizer that included *Potentilla anserina* extract on the skin inside and outside the mask. All 33 subjects completed the trial without dropouts. Regardless of the presence of the *Potentilla anserina* extract, the skin redness significantly decreased after applying moisturizer after 2 and 4 weeks (*p* < 0.05) (Figure 4A). This was consistent with the report that skin redness is improved by moisturizers [10]. The hemoglobin values (A.U.), which indicate skin redness, decreased by 5.06% and 6.74% in the test area inside the mask after 2 and 4 weeks, respectively, compared to before the trial, while the control area inside the mask decreased by 5.73% and 6.05% after 2 and 4 weeks, respectively. Based on these results, the use of moisturizers was deemed to be effective in reducing skin redness over time, regardless of the presence of *Potentilla anserina* extract.

### 2.8. Skin Barrier Improvement Effects

Using masks increased the TEWL, especially inside the mask compared to outside. As the time progressed from 0 weeks to 4 weeks, the moisturizer containing the *Potentilla anserina* extract significantly reduced the TEWL (Figure 4B, treated). The TEWL values decreased by 5.23% and 9.13% in the test area inside the mask after 2 and 4 weeks, respectively, and by 13.63% and 16.62% in the test area outside the mask after 2 and 4 weeks, respectively, compared to before the trial. One important point to note is that, in the control group where the *Potentilla anserina* extract was not applied, there was no observed tendency of the TEWL to increase or decrease over time (Figure 4B, non-treated). In this area, the TEWL inside the mask decreased by 0.04% after 2 weeks and increased by 2.88% after 4 weeks, while the TEWL outside the mask decreased by 3.92% and 7.08% after 2 and 4 weeks, respectively.

The fact that only the experimental group with the *Potentilla anserina* extract showed an improvement in TEWL over time implies that it strengthened the skin barrier. This aligns with the inherent Filaggrin-enhancing effects of *Potentilla anserina* extract.

### 2.9. Skin Moisture Improvement Effects

When a moisturizer containing *Potentilla anserina* extract was applied, a significant increase in moisturizing efficacy over time was observed (Figure 4C, treated). On the other hand, moisturizers without the *Potentilla anserina* extract showed limited moisturizing effects (Figure 4C, non-treated). The Corneometer values (A.U) increased by 11.51% and 15.14% in the test area inside the mask after 2 and 4 weeks, respectively, and by 13.51% and 24.83% in the test area outside the mask after 2 and 4 weeks, respectively, compared to before the trial. In the control area, the Corneometer values inside the mask increased by 3.13% and 3.92% after 2 and 4 weeks, respectively. The *Potentilla anserina* extract increased AQP3, which is consistent with these Corneometer values.

## 3. Discussion

Our skin is crucial to shielding us from external factors and keeping our body’s internal balance stable. However, a growing number of individuals have reported skin issues associated with wearing masks amid the ongoing pandemic [1,4,9]. Research into skin sensitivity based on race has revealed a relatively high incidence of sensitive skin among Korean women [32]. Consequently, the pandemic and its mask-wearing practices have prompted interest in understanding and treating the mask-induced changes in this population.

The ongoing COVID-19 pandemic has no immediate cure, and even with vaccinations, people worldwide continue to become infected. As a result, many localities have implemented mask mandates, particularly with the emergence of new SARS-CoV-2 variants [2,33]. However, the prolonged use of masks has been reported to cause adverse dermatological changes. This is not just a problem of COVID-19. There is always a possibility of a similar infectious disease starting at any time, since individuals with sensitive skin experience negative dermatological effects from prolonged mask wearing.

The aim of this study was to confirm the safety and absence of side effects of a moisturizer containing *Potentilla anserina* extract encapsulated in HPβCD-Liposome for gradual release. One of the major substances (caffeic acid) of *Potentilla anserina* extract was encapsulated in the liposomes, leading to a sustained release. The liposomes enhanced the skin permeation of the caffeic acid, with HPβCD helping.

In this study, the Drug-in-cyclodextrin-in-liposome (DCL) system used was specialized for stabilizing drugs and controlling their release. In particular, the DCL system is effective in capturing hydrophobic or hydrophilic components. The efficacious components of flavonoids, including caffeic acid, are generally easily captured by cyclodextrins [34]. The main efficacious components of *Potentilla anserina* extract, including caffeic acid, were primarily captured by HPβCD and subsequently encapsulated in liposomes. It was expected that the DCL system would slow down the release rate, reducing the irritation of the *Potentilla anserina* extract. Furthermore, it helped to maintain efficacy through continuous release. In particular, the liposomes enhanced the skin permeation of the caffeic acid, with HPβCD aiding in this process. From these results, it was expected that the effective substances of this extract would gradually be released into troubled areas of the skin wearing the mask.

Wearing a mask increases the skin redness and TEWL of sensitive skin, and these changes are thought to be caused by physical friction between the mask material and the skin, as well as an elevated temperature in the perioral area [32,35]. Applying the moisturizer after mask wearing significantly improved the skin redness and TEWL, and these findings indicated that using moisturizers periodically on sensitive skin could prevent the temporary redness or dryness caused by mask wearing. Redness and dryness cause problems with wrinkles and pores in the skin. This is connected to the fact that using moisturizers during prolonged mask wearing can be effective in improving wrinkles and pores [2,5].

We attempted to develop a moisturizer containing specific ingredients, rather than simply using a basic moisturizer. To do this, we focused on *Potentilla anserina* extract. The *Potentilla anserina* extract has been tested for its effects on skin hydration, skin barrier improvement, and itching relief efficacy. The experiment showed that the extract increased the expression of the AQP3 protein, which affects skin hydration, and the expression of the FLG protein, which improves the skin barrier. The extract also reduced the expressions of cytokines such as TSLP, IL-33, and IL-4, which are commonly expressed in atopic dermatitis and play roles in causing skin itching, suggesting that the extract may have had a notable effect on reducing itching.

Irrespective of the presence of the *Potentilla anserina* extract, the application of a moisturizer elicited a substantial reduction in cutaneous erythema after 2 and 4 weeks, consistent with the prior literature highlighting the erythema-alleviating properties of moisturizers [10]. Masks were determined to augment the TEWL, particularly in the regions covered by the mask compared to the exposed skin. Over the four-week timeframe, the application of a moisturizer enriched with *Potentilla anserina* extract led to a significant attenuation of the TEWL. It is noteworthy that, within the control group, wherein the *Potentilla anserina* extract was absent, no discernible temporal trend in TEWL perturbations was observed. The exclusive amelioration of the TEWL discerned in the experimental group treated with the *Potentilla anserina* extract implied its augmentation of the skin’s epidermal barrier, aligning with the established Filaggrin-enhancing attributes of *Potentilla anserina* extract.

Upon the application of a moisturizer containing *Potentilla anserina* extract, a substantial and time-dependent enhancement in cutaneous hydration was observed. Conversely, the moisturizers devoid of the *Potentilla anserina* extract exhibited limited hydrating effects. This investigation unequivocally substantiates the substantial enhancement in the skin hydration levels conferred by formulations incorporating *Potentilla anserina* extract after 2 and 4 weeks, whereas their non-inclusion failed to produce commensurate results. In simpler terms, the conventional moisturizers did not yield analogous levels of cutaneous hydration. The synergistic effect between the extract and the moisturizer manifested promising outcomes, corroborated by its association with an augmented AQP3 expression, as affirmed by the Corneometer measurements.

These findings underscore the need for advanced formulations beyond conventional moisturizers. Particularly intriguing was the augmentation of moisturizing efficacy upon the inclusion of this extract. Moreover, the remarkable improvement in TEWL mitigation achieved by the extract-enriched moisturizer was noteworthy. The development of such enhanced function moisturizers holds promise in providing robust protection, even in the face of unforeseen events, such as the advent of COVID-19.

## 4. Materials and Methods

### 4.1. Preparation of Potentilla anserina Extract and Analysis Conditions Using HPLC

The preparation of the *Potentilla anserina* extract involved a multi-step process. It began with drying the original plant material, which was then ground into a fine powder. This powder was soaked in distilled water at a ratio of 1:10 (*w*/*v*), meaning that 100 g of dried plant material was soaked in 1000 mL of distilled water. The mixture was then heated to 121 °C for 15 min to facilitate extraction. Once the extraction was complete, the liquid extract was initially filtered through a mesh to remove any solid particles. Following this filtration step, the remaining extract was subjected to freeze drying to obtain a solid extract. To further refine this solid extract, it was dissolved in 70% alcohol to remove any precipitated debris. This solution was then passed through a 0.2 μm filter to ensure its purity. After this purification process, a second round of freeze drying was performed to obtain the final solid extract, which was used in the experiment.

The equipment used to determine the presence of caffeic acid in the *Potentilla anserina* extract was the Thermo-Finnigan Surveyor instrument (Thermo Scientific, Waltham, MA, USA). The Shim-pack VP-ODS C18 column (L250 mm LD: 4.6 mm, 5 μm) was utilized as the column and the UVD 170 s DIONEX was used as the measurement detector, with measurements being taken at a wavelength of 324 nm. For the mobile phase, A consisted of 2% acetic acid and B consisted of 0.5% AA in 50% acetonitrile. Isocratic conditions were employed without gradient changes. The mass spectrometric analysis equipment used to confirm the target fraction was the Thermo Finnigan LCQ Deca XP plus ion trap mass spectrometer with an ESI interface.

### 4.2. Cell Culture and Condition

To culture the human keratinocyte-forming cells (HaCaT) and Detroit cells (human fibroblasts, Detroit 551), DMEM supplemented with 10% FBS and 1% antibiotic-antimycotic (GIBCO, Billings, MT, USA) was used. The cells were grown under conditions of 37 °C and 5% CO_2_.

### 4.3. Real-Time Quantitative Reverse Transcriptase Polymerase Chain Reaction (qRT-PCR)

In the experiment, a real-time quantitative reverse transcriptase polymerase chain reaction (qRT-PCR) was employed to measure the expression levels of filaggrin (FLG), aquaporin-3 (AQP3), and cytokine mRNA in the cells. RNA isolation was performed using the FastLane Cell one-step buffer set from QUAGEN, and cDNA synthesis was carried out using the 2x QuantiTect (QUAGEN, Venlo, The Netherlands) SYBR Green RT-PCR Master mix. The real-time PCR analysis utilized the Thunderbird™ (TOYOBO, Osaka, Japan) SYBR qPCR Mix. QuantiTect primer assays were used as primers (AQP3; Cat. QT00212996, FLG; Cat. QT01192646, IL-4; Cat. QT00012565, TSLP; Cat. QT00051464, IL-33; and Cat. QT00041559), with GAPDH serving as the reference gene (GAPDH; Cat. QT01192646). The real-time cycler conditions for this experiment included a 30 min reverse transcription step at 50 °C, followed by a 15 min PCR initial activation step at 95 °C. The cycling phase consisted of 45 cycles, each comprising a 15 s denaturation step at 94 °C, a 30 s annealing step at 60 °C, and a 30 s extension step at 72 °C. The expression rate was calculated using the formula: Expression rate = (expression rate of treated sample)/(expression rate of control sample).

### 4.4. Preparation of Hydroxypropyl-β-Cyclodextrin (HPβCD)-Liposome

Hydroxypropyl-β-cyclodextrin (HPβCD) was utilized to encapsulate the *Potentilla anserina* extract or its major component, caffeic acid. The resulting complex was then incorporated into liposomes, which were referred to as HPβCD-Liposomes [36]. The caffeic acid and HPβCD were mixed in a 1:1 ratio and dissolved in 50% ethanol at room temperature. The *Potentilla anserina* extract was prepared following a method like that of the production of caffeic acid. After stirring, the solution underwent vacuum distillation to obtain a powdered complex. To encapsulate this complex, an aqueous phase containing lecithin and water was heated, and the lipid phase was added while homogenizing.

### 4.5. Particle Size and Zeta Potential Evaluation

The liposomes’ particle sizes and distributions were analyzed using dynamic light scattering at 25 °C [37], with measurements being taken three times with 70 runs per measurement. The average particle size and zeta potential were determined using the cumulative analysis method and were measured three times with 10 runs per measurement, respectively.

### 4.6. Determination of Entrapment Efficiency (EE%) of Caffeic Acid

Spectrophotometry was used to measure the entrapment efficiency (EE%) of the caffeic acid (CA) in the liposomes and HP-β-CD-Liposomes [38]. After centrifugation, the remaining amount of CA in the supernatant was measured using a UV-VIS spectrophotometer. The concentration was determined using a calibration curve obtained with the CA. The EE% was calculated using the following equation.
$$\mathrm{EE}\left(\%\right)=\frac{\mathrm{Tca}\;-\mathrm{Ts}}{\mathrm{Ts}}\times 100$$
where Tca represents the total amount of CA and Ts represents the amount of CA in the supernatant.

### 4.7. Release Profile Evaluation and Skin Permeation of Caffeic Acid

The liposome release was assessed using dialysis with a 1 mL liposome solution in a pre-soaked dialysis bag immersed in 30 mL of release media. In vitro skin permeation experiments were conducted using Franz diffusion cells [37], where the skin was placed between the donor and receptor phases. The receptor chamber contained 5 mL of the receptor phase at 36 °C. Each formulation (0.2 mL) was applied to the skin, and receptor-phase samples were collected and replenished. The CA concentration was measured using UV-visible spectrometry.

### 4.8. Particle Size and Zeta Potential Evaluation

The liposomes’ particle sizes and distributions were analyzed using dynamic light scattering at 25 °C [37], with measurements being taken three times with 70 runs per measurement. The average particle size and zeta potential were determined using the cumulative analysis method and were measured three times with 10 runs per measurement, respectively.

### 4.9. Instrument-Based Clinical Measurements

#### 4.9.1. Ethical Approval

This study was granted approval by the Institutional Review Board of the Korea Dermatology Research Institute (KDRI-IRB-230106-A). The research followed the guidelines set forth in the Declaration of Helsinki and the Good Clinical Practice (GCP) in conducting research and recording results.

#### 4.9.2. Skin Redness

ANTERA 3D (Miravex, Dublin, Ireland) was used to measure the skin redness of the target areas. This equipment is a device that analyzes the skin’s condition by imaging the reflected signal when illuminating the skin, with visible light of different frequencies being used as a light source. The hemoglobin values (A.U.), which indicate skin redness, were measured according to the device manual.

#### 4.9.3. Transepidermal Water Loss (TEWL)

The measurement of the TEWL was performed using a Tewameter TM300 (Courage & Khazaka GmbH, Cologne, Germany) placed on the targeted areas. The device determined the density gradient of the water evaporating from the skin, expressed in g/h/m^2^.

### 4.10. Subject Recruitment

#### 4.10.1. Moisturizer

The moisturizer titled Celladerm Youth Turning Ampoule (Kolmar Korea, Seoul, Republic of Korea), which contains *Potentilla anserina* extract encapsulated in HPβCD-Liposome was used in this study. The composition included fundamental dosage form components, including water (up to 100%), glycerin (6%), caprylic/capric triglyceride (4%), niacinamide (2%), 1,2-hexanediol (2%), and panthenol (1%). It also incorporated HPβCD-Liposome-related ingredients such as hydrogenated lecithin (2%), sucrose distearate (0.01%), hydroxypropyl cyclodextrin (0.10%), and *Potentilla anserina* extract (0.05%), alongside minimal amounts of preservatives (sodium benzoate, hydroxyacetophenone, and ethylhexylglycerin), a diverse range of natural origin elements, and supplementary additives. The key distinction between the treated and non-treated (control) groups was the presence of the *Potentilla anserina* extract, with the former containing it while the latter did not, with all other components remaining consistent.

#### 4.10.2. Applying Moisturizer

The study applied a moisturizer to the faces of men and women aged from 19 to 70 who wore masks for more than 6 h a day. The faces were divided into a test area (treated) and a control area (non-treated), with measurements being taken inside and outside of the mask. In the treated group, a moisturizer containing *Potentilla anserina* extract was applied, while, in the non-treated group, a moisturizer without *Potentilla anserina* extract was applied. The only difference between the two moisturizers was the presence or absence of the *Potentilla anserina* extract, with all the other formulations being identical. A flowchart of the clinical trials used in this study is presented in Figure 5.

The moisturizer was applied twice a day, and the skin hydration, TEWL, and skin redness were checked at 0, 2, and 4 weeks. The study used three different methods to measure the effects on the skin: TEWL using a Tewameter, skin moisture using a Corneometer CM 825 (Courage & Khazaka GmbH, Cologne, Germany), and skin redness using an Antera 3D. The improvement rates were calculated using specific formulas for each method, and measurements were taken before the test and 2 and 4 weeks after. The improvement rate formula (Hemoglobin/TEWL value) was as follows. Hemoglobin/TEWL value = Average [(post-test measurement from each subject - pre-test measurement from each subject)/(pre-test measurement from each subject)] × 100.

In the case of skin moisture (Corneometer value), the formula was as follows. Corneometer value = Average [(post-test measurement from each subject—pre-test measurement from each subject)/(pre-test measurement from each subject)] × 100.

#### 4.10.3. Statistical Analysis

The statistical analysis was carried out using the SPSS software version 28.0 (IBM Corp., Armonk, NY, USA), with results being presented as mean ± standard deviation. The normality was evaluated using the Shapiro–Wilk test, with a significance level of *p*-value > 0.05. The comparison between the various conditions was performed using a repeated measures analysis of variance, and a *p*-value of <0.05 was considered to be statistically significant.

## 5. Conclusions

Our study confirmed the effectiveness of using moisturizers for improving the side effects of wearing masks, and we found that the formulated moisturizer could provide even better results. We also discovered that the regular use of moisturizers could prevent the temporary redness or dryness caused by mask wearing. Furthermore, the sustained release system of HPβCD-Liposomes could reduce irritation and minimize side effects, making it highly suitable for sensitive skin that has become more sensitive due to wearing masks [30]. To prevent the resurgence of a possible future pandemic, the continuous development of moisturizers that improve the side effects of mask wearing is necessary. Above all, it was evident that the moisturizers fortified with specific ingredients were more helpful for addressing the skin issues caused by mask irritation, rather than simple moisturizers. Our research implied that moisturizers with added specific components outperform simple moisturizer treatments. Therefore, we look forward to ongoing research related to this before COVID-19 or similar respiratory diseases surge again, hoping that the public and healthcare professionals will no longer have to endure the discomfort caused by masks. We also anticipate similar follow-up studies in this regard.

## Figures and Tables

**Figure 1 ijms-24-14294-f001:**
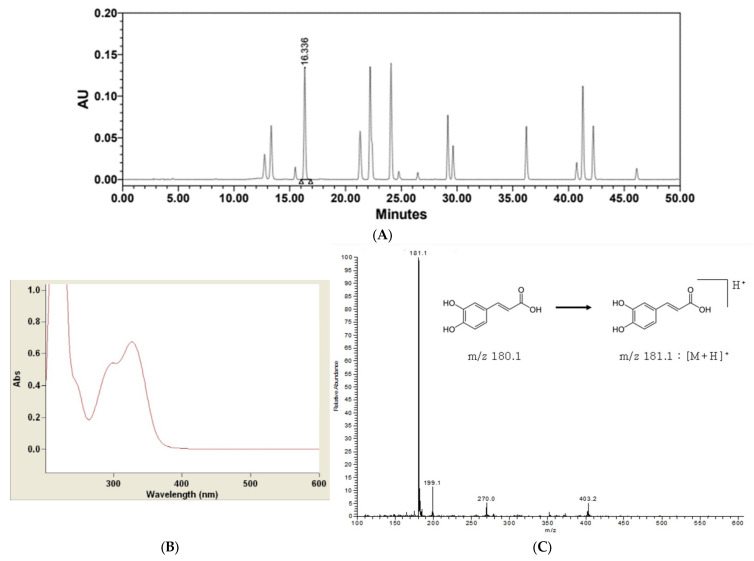
HPLC data from *Potentilla anserina* extract (**A**). UV scanning of the fraction at 16.336 min (**B**), and LC-MS data of the fraction at 16.336 min (**C**).

**Figure 2 ijms-24-14294-f002:**
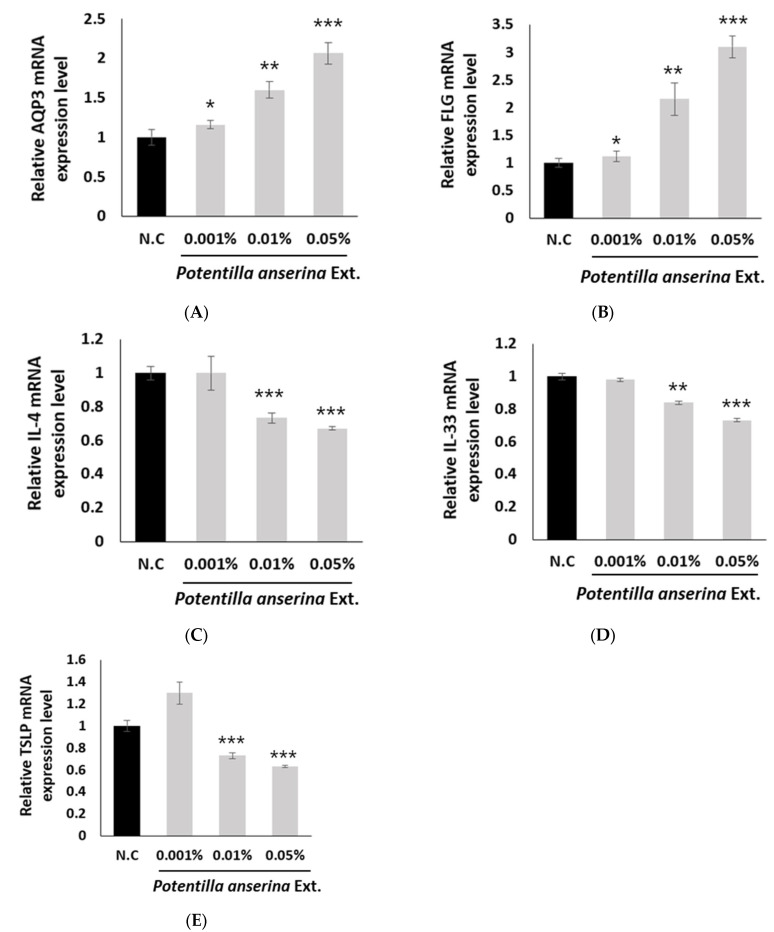
The effects of *Potentilla anserina* Extract on (**A**) AQP3, (**B**) FLG, (**C**) IL-4, (**D**) IL-33, and (**E**) TSLP in vitro model using HaCaT cells. AQP3: Aquaporin-3/FLG: Filaggrin/IL-4: Interleukin-4/IL-33: Interleukin-33/TSLP: Thymic Stromal Lymphopoietin/N.C: (Negative) Control. Significant differences were found when comparing with N.C: * *p* < 0.05, ** *p* < 0.01, and *** *p* < 0.001.

**Figure 3 ijms-24-14294-f003:**
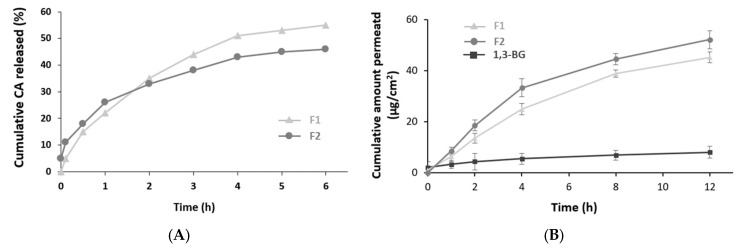
(**A**) The release profiles of CA from liposomal formulations were quantitatively analyzed and graphed. (**B**) Skin permeation profiles of CA, whether dissolved in 1,3-butylene glycol solution (1,3-BG), CA/LIP (F1), or CA/HPβCD/LIP (F2), were examined in vitro over a 12-h period.

**Figure 4 ijms-24-14294-f004:**
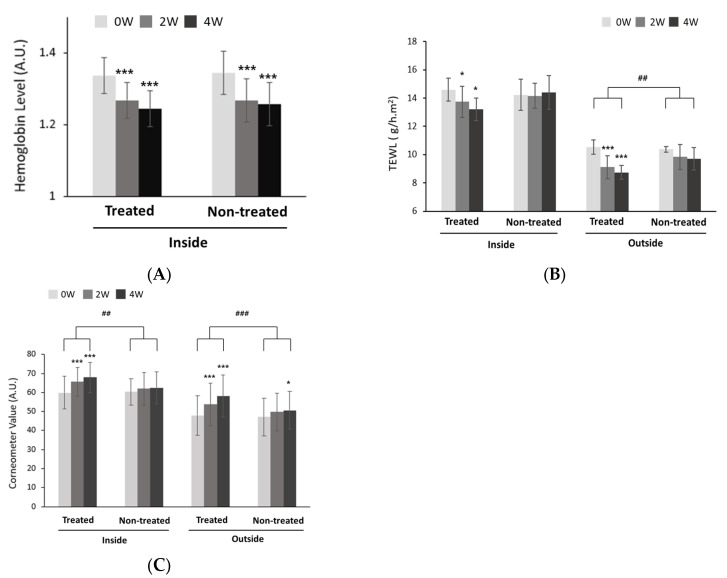
Efficacy and effects of moisturizer including *Potentilla anserina* extract in a clinical trial model. Treated group: subjects who applied a moisturizer containing *Potentilla anserina* extract. Non-treated group: subjects who applied a control moisturizer without *Potentilla anserina* extract., (**A**) Hemoglobin level (A.U.), (**B**) TEWL (g/h/m^2^), and (**C**) Coneometer value, significant probability within group: * *p* < 0.05, and *** *p* < 0.001, significant probability between group: ## *p* < 0.01, and ### *p* < 0.001.

**Figure 5 ijms-24-14294-f005:**
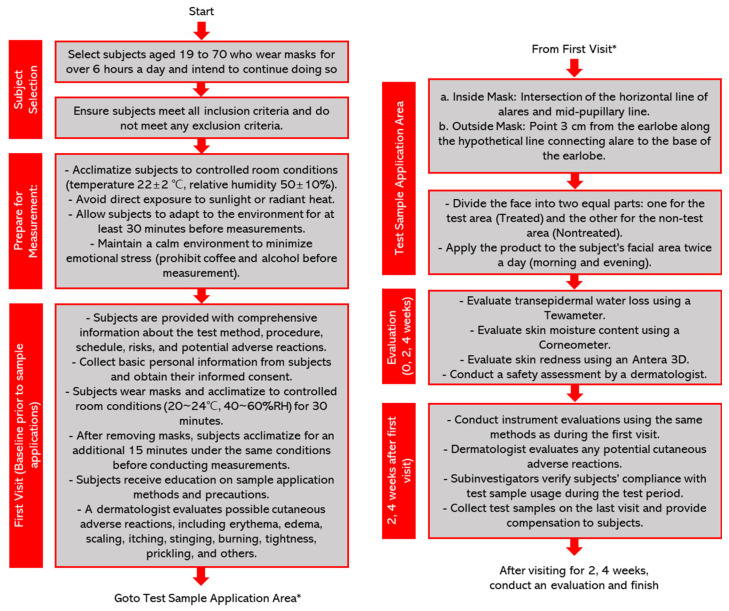
Flowchart of the clinical trials used in this study. The flowchart progresses from * at the bottom to * at the top.

**Table 1 ijms-24-14294-t001:** Liposome characteristics with caffeic acid and/or hydroxypropyl-β-cyclodextrin. LIP: liposome/CA: caffeic acid/HPβCD: hydroxypropyl-β-cyclodextrin. Significant probability within F0: * *p* < 0.05, significant probability within F1: # *p* < 0.05.

Formulation	Particle Size (nm)	Polydispersity Index	Zeta Potential (mV)	Entrapment Efficiency of CA (%)
F0. LIP	195.2 ± 5.2	0.178	−28.2 ± 0.3	-
F1. LIP/CA	201.3 ± 4.2 *	0.166	−32.1 ± 0.1 *	82.5 ± 1.5
F2. LIP/CA/HPβCD	215.2 ± 8.2 *^,#^	0.195	−34.2 ± 0.5 *^,#^	75.6 ± 3.1

## Data Availability

The data that support the findings of this study are available from the corresponding author upon reasonable request.

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
