# Peer review of "The Protective Effects of Moisturizer Containing Potentilla anserina Extract in the Topical Treatment of Skin Damage Caused by Masks"

_ijms, 2023, doi:10.3390/ijms241814294_

Round 1

Reviewer 1 Report

 The manuscript present important findings; however, some comments should be addressed as follows:  

1.     Abstract: it should be strengthened by adding quantitative data.

2.     Introduction: the novelty of this study should be elucidated. The aim should be elaborated at the end.

3.     Line 66: could should be corrected.

4.     I presume that the extract included other phenolic compounds in addition to Caffeic Acid, which could have antioxidant and anti-inflammatory properties. Could you explain? Have you done HPLC besides LC-MS using standard flavonoids? For instance, please check this article. https://doi.org/10.3390/catal13020443.

5.     Fig. 1 c should be improved.

6.     Statistical analysis in Table 1 should be added.

7.     Discussion: it is too short. The authors should elaborate the discussion section in the light of the extensive results obtained in this study, comparing their results with previous reports.

8.     Methodology: which primer did you use for qRT-PCR? Please add the nucleotide sequences of the primers and the conditions applied for cDNA synthesis and qRT-PCR.

Conclusion: the authors should highlight the significant findings and the future perspectives precisely (if possible). 

Author Response

I want to express my heartfelt gratitude for your thoughtful and dedicated participation in the paper review process. Your thorough review was highly valuable, and I have made revisions in response to most of your inquiries.

Once more, I would like to convey my deep appreciation for your invaluable input, which has significantly enhanced the manuscript. These revisions aim to bring the manuscript more in line with your expectations, and we eagerly anticipate your ongoing guidance and feedback as we progress with our research.

Thank you for your valuable time and unwavering commitment to advancing our research.

  1. Abstract : it should be strengthened by adding quantitative data.

: Thank you for the valuable feedback. I have added quantitative data in lines 15-19 and 22-32 as suggested.

  1. Introduction: the novelty of this study should be elucidated. The aim should be elaborated at the end

: I have mentioned the reasons for the specific ingredient enhancement in moisturizers as in lines 67-74 and clarified the experimental objectives as in lines 93-96.

  1. Line 66: Could should be corrected.

: I have made the revisions as suggested in line 80.

  1. I presume that the extract included other phenolic compounds in addition to Caffeic Acid, which could have antioxidant and anti-inflammatory properties. Could you explain? Have you done HPLC besides LC-MS using standard flavonoids? For instance, please check this article. https://doi.org/10.3390/catal13020443.

: The primary objective of this study is to alleviate skin irritation caused by masks using this extract, and we have determined that substance identification is not the primary focus. Since other known acids and flavonoid components from previous research have been confirmed, https://www.ncbi.nlm.nih.gov/pmc/articles/PMC6272682/, we did not emphasize novelty beyond the indicators. We kindly request your understanding and consideration of this aspect.

  1. Fig. 1 c should be improved.

: Thank you for your valuable feedback. I have improved the images as suggested.

  1. Statistical analysis in Table 1 should be added.

: Thank you for your feedback. I have incorporated the requested changes.

  1. Discussion: it is too short. The authors should elaborate the discussion section in the light of the extensive results obtained in this study, comparing their results with previous reports.

: I have extensively enhanced the sections highlighted in red. Thank you once again for your feedback.

  1. Methodology: which primer did you use for qRT-PCR? Please add the nucleotide sequences of the primers and the conditions applied for cDNA synthesis and qRT-PCR.

: I have included the PCR conditions and cDNA information. Since these are specific primers from a particular company, I have included the catalog numbers of the primers instead of the sequence. Thank you for your feedback.

  1. Conclusion: the authors should highlight the significant findings and the future perspectives precisely (if possible).

: I sincerely appreciate your overall feedback on the Conclusion section. Considering your suggestions, I have emphasized and elaborated on the outlook and expectations. Thank you for your valuable input.

Reviewer 2 Report

A very nice work

Regarding the in vivo experiments to human volunteers (TEWL, redness some points are missing )

1) Flowchart of the clincal trials

2) The ingredients of the forlulation-moisturizer mask

3) Placebo?

As I can understand there was only control 

4) Additionally you have to express more precisely a) there wre two groups A treated b) B not treated or 

In the same group the tested product was applied on one area (of the face?) ant the other area was not treated?

Since, no placebo was used, I think that the ingredients and their concentration  in the mask have to be mentioned. 

The title synergetic is not appropriate. 

Author Response

I am grateful for your thoughtful feedback and kind words. Your meticulous review of the paper has been invaluable. I have incorporated the majority of the requested revisions and addressed your inquiries.

I want to reiterate my sincere appreciation for your constructive input, which has significantly improved the manuscript. These revisions aim to better align the paper with your expectations, and we eagerly anticipate your ongoing guidance and feedback as we progress with our research.

Thank you once again for your dedicated time and commitment to advancing our work.

  1. Flowchart of the clinical trials

: You're welcome. I have attached Figure 5. Thank you.

  1. The ingredients of the formulation-moisturizer mask

: I have included an attachment for the cosmetic formulations used as described in lines 418-426. Thank you.

  1. Placebo?

: Firstly, it may be helpful to understand the term 'placebo' as 'Non-treated (moisturizer)' in the context of this experiment. The distinction between a simple moisturizer and a placebo has been made based on existing research results. We assume that moisturizers have an effect, and our experiment's main objective is to investigate the time-dependent differences between moisturizers with and without the extract. We believe this is essential to evaluate.

  1. As I can understand there was only control: Understanding 'control' as 'Non-treated' should suffice for clarification. Additionally, you must express more precisely a) there were two groups A treated b) B not treated or In the same group the tested product was applied on one area (of the face?) ant the other area was not treated?

: Apologies for any confusion in the paper's description, and thank you for the clarification. To prevent any further confusion, I have strengthened the labeling in the results section, specifically in lines 224, 232, and 234. Additionally, I have provided a more detailed explanation from lines 431 to 437 for added clarity.

  1. Since, no placebo was used, I think that the ingredients and their concentration in the mask have to be mentioned.

: I have included the formulation details of the cosmetic preparations used as described in lines 418-427. Thank you for providing this information.

  1. The title synergetic is not appropriate.

: I have taken your feedback into consideration and made revisions to the title.

Reviewer 3 Report

The subject of the article is consistent with the current trends on the cosmetics market regarding the search for safe, mild and effective cosmetic ingredients. Due to the use of face masks during the COVID-19 pandemic, numerous skin irritations or their aggravation have been reported, especially for people with sensitive skin. Wearing masks for a long time can result in skin problems such as acne, contact dermatitis, erythema, papules and pustules on the face. The study examined the effect of the Potentilla anserina extract on, among others, skin hydration, barrier function and itching. The literature presents research results showing that moisturizers can alleviate irritation caused by masks, however, there is little information about research that would improve not only irritation, but also itching and the skin barrier. There are reports that long-term use of masks weakens the skin barrier, and people who wear masks for a long time are prone to more itching sensations. However, little is known about the specific substances and methods other than simply applying a moisturizer to the skin.

Therefore, taking up this topic is justified and contains elements of novelty.

Nevertheless, the following shortcomings were noticed during reading:

• L.2 p.70 - it would be good to write here what kind of extract it was

• It is worth providing information in the introduction whether the extract of Potentilla anserina has already been applied to the skin, eg in moisturizing creams? Is this new?

• In the introduction, the general composition of this extract should be given (from studies by various authors)

• When presenting the research results, the authors should write what the extract was

• Section 2.7. p.8 - write the composition of the cream according to INCI

Author Response

Your kind words are greatly appreciated. I would like to express my sincere gratitude for your thorough review of the paper. I have implemented most of the requested revisions and have addressed your inquiries.

Once again, I want to convey my appreciation for your invaluable feedback, which has significantly improved the manuscript. We believe that these revisions bring the manuscript in closer alignment with your expectations, and we eagerly anticipate your ongoing guidance and feedback as we proceed with our research endeavors. Thank you for your dedicated time and commitment to advancing our research.

  1. L.2 p.70 - it would be good to write here what kind of extract it was

: To clarify the nature of the extract, I have enhanced section 4.1. 'Preparation of Potentilla anserina extract' in the 'Materials and Methods' to explicitly describe what this extract is. If there are other specific areas you would like me to address, please let me know.

  1. It is worth providing information in the introduction whether the extract of Potentilla anserina has already been applied to the skin, eg in moisturizing creams? Is this new?

: As you suggested, I have elaborated on the description in lines 84-96. I emphasized that while there is literature related to the genus Potentilla and its relevance to skin, there are relatively few instances of Potentilla anserina being used in cosmetics applications. While there is a history of this extract being used as an astringent in cosmetics, scientific reports related to this usage are scarce.

  1. In the introduction, the general composition of this extract should be given (from studies by various authors)

: Potentilla erecta and especially Potentilla paradoxa Nutt., which belong to the Potentilla genus, indeed have a strong connection with the skin. However, despite being a part of the Potentilla genus, Potentilla anserina's research related to the skin is relatively sparse. This scarcity of research is what motivated us to plan and conduct this study.

  1. When presenting the research results, the authors should write what the extract was

: I have restructured the introduction section, particularly the part about Potentilla anserina, and provided a clearer description of the extraction method from line 323 onwards. Thank you for your efforts in improving the manuscript.

  1. Section 2.7. p.8 - write the composition of the cream according to INCI

: I have included the formulation details of the cosmetic preparations used as described in lines 418-427. Thank you for providing this information.

Round 2

Reviewer 1 Report

The authors have successfully addressed the reviewers' comments, made substantial revisions, and improved the overall quality and clarity of the manuscript.

Reviewer 2 Report

Compliance with my comments